# Regional Potential Wind Erosion Simulation Using Different Models in the Agro-Pastoral Ecotone of Northern China

**DOI:** 10.3390/ijerph19159538

**Published:** 2022-08-03

**Authors:** Jun Liu, Xuyang Wang, Li Zhang, Zhongling Guo, Chunping Chang, Heqiang Du, Haibing Wang, Rende Wang, Jifeng Li, Qing Li

**Affiliations:** 1College of Resource and Environment Sciences/Hebei Key Laboratory of Environmental Change and Ecological Construction, Hebei Normal University, Shijiazhuang 050024, China; lj9499lj@163.com (J.L.); wangqq158111@163.com (X.W.); zhangli210826@163.com (L.Z.); ljfgis@163.com (J.L.); 2Northwest Institute of Eco-Environment and Resources, Chinese Academy of Sciences, Lanzhou 730000, China; duheqiang@lzb.ac.cn; 3College of Desert Control and Science and Engineering, Inner Mongolia Agricultural University, Huhhot 010018, China; hbwang@lzb.ac.cn; 4Institute of Geographical Sciences, Hebei Academy Sciences/Hebei Engineering Research Center for Geographic Information Application, Shijiazhuang 050011, China; wangrende10@163.com (R.W.); qingli2020@outlook.com (Q.L.)

**Keywords:** potential wind erosion, wind erosion model, calibration, validation, comparison

## Abstract

Wind erosion is crucial for assessing regional ecosystem services and sustainable development. The Agro-Pastoral Ecotone of northern China (APEC) is a typical region undergoing wind erosion and soil degradation. In this study, the National Wind Erosion Survey Model of China, the Integrated Wind Erosion Modeling System, and the regional versions of the Revised Wind Erosion Equation and Wind Erosion Prediction System were used to evaluate the regional potential wind erosion of the APEC during 2000 and 2012. The results showed that the potential wind erosion predicted by National Wind Erosion Survey Model of China (NWESMC), Revised Wind Erosion Equation (RWEQ), Wind Erosion Prediction System (WEPS), and Integrated Wind Erosion Modeling System (IWEMS) were significantly related to the observed wind erosion collected from published literature, but the observed data were generally smaller than the predicted values. The average potential wind erosions were 12.58, 25.87, 52.63, and 58.72 t hm^−2^ a^−1^ for NWESMC, RWEQ, WEPS, and IWEMS, respectively, while the spatial pattern and temporal trend of annual potential wind erosion were similar for different wind erosion models. Wind speed, soil moisture, and vegetation coverage were the dominant factors affecting regional wind erosion estimation. These results highlight that it is necessary to comprehensively calibrate and validate the selected wind erosion models. A long-term standard wind erosion monitoring network is urgently required. This study can serve as a useful reference for improving wind erosion models.

## 1. Introduction

From most arid and semi-arid regions to some high-latitude and high-altitude areas, wind erosion is one of the main processes causing soil loss and dust emission, which further results in soil degradation and reductions in land productivity [1,2,3,4]. Chappell [5] called on the academic community to consider and deal with wind erosion and dust emissions more widely. Onsite direct observation [6,7,8], rare earth elements or radio isotope (such as 137Cs, 7Be) surveys [9,10], and wind erosion modeling [11,12] were generally summarized as the main methods to quantitatively determine wind erosion. Of those, wind erosion models were considered the most efficient method to obtain regional patterns of wind erosion [13]. Many wind erosion models were developed to evaluate wind erosion at different scales for different land use [14,15].

Based primarily on United States Department of Agriculture research, Woodruff and Siddoway [16] first proposed a wind erosion model named the Wind Erosion Equation (WEQ). Subsequently, the Revised Wind Erosion Equation (RWEQ) [11], the Wind Erosion Predicted System (WEPS) [17], the Texas Tech Erosion Analysis Model (TEAM) [18], the Wind Erosion Stochastic Simulator (WESS) [19], and other models were developed. Originally, these models were designed to evaluate wind erosion of farmlands at a field scale. Furthermore, these models were extensively validated, revised, or intercompared with various environments at field scale [20,21,22,23,24]. To meet the challenge of precisely estimating wind erosion with different land use at a regional scale, some field-scale wind erosion models, such as RWEQ and WEPS, have been scaled up to regional versions with different methodology for specific purposes [25,26,27,28]. Moreover, wind erosion models with regional scale were also constructed. These models included the Wind Erosion Assessment Model (WEAM) [29], the Integrated Wind Erosion Modeling System (IWEMS) [12], the AUStralian Land Erodibility Model (AUSLEM) [14], the National Wind Erosion Survey Model of China (NWESMC) [30], the Aeolian EROsion (AERO) model [31], and so on. These models incorporated remote sensing image and weather data with geographic information system data and were widely used to assess the spatio-temporal pattern of regional wind erosion under different environments across different countries [27,32,33]. The magnitudes of regional wind erosion determined by these models were further used to conduct the effectiveness assessment of ecosystem services [34].

From water erosion models to dust emission schemes, it has been demonstrated that different water erosion or dust emission models generally produced different simulated results [35,36]. It is unclear whether different wind erosion models could yield different regional wind erosion estimates for the same target modeling region based on the same dataset. However, there have been no comparisons of popular models for regional wind erosion modeling. The purposes of this study were: (1) to estimate the spatial-temporal trends of potential wind erosion using the regional versions of RWEQ and WEPS together with IWEMS and NWESMC in the APEC, (2) further validate the evaluated potential wind erosion using observed wind erosion data, and (3) to investigate the main factors affecting the regional potential wind erosion.

## 2. Materials and Methods

### 2.1. Study Area

APEC refers to the transitional zone between the monsoon region in eastern China and the arid and semi-arid region in northwest China. Although the detailed boundaries of APEC might differ for different research purposes, the core areas are roughly the same [37]. In this study, the boundary described by Guo [32] was used to define the geographical scope of APEC. The study area is approximately 36°30′ N–46°42′ N, 106°16′ E–124°51′ E (Figure 1), including Inner Mongolia, Liaoning, Jilin, Hebei, Shanxi, and Shaanxi provinces, covering an area of 543,616 km^2^. The land use types in the study area are various, mainly including arable land, forest land, grass land, and sand land. Most of the research area is semi-arid, with an average annual precipitation of 300–450 mm. The precipitation mainly occurs in summer and autumn. The average wind speed in this area is 2.49 m s^−1^, and the maximum wind speed is 16–24 m s^−1^. Because of the sensitivity of the ecological environment, desertification has developed rapidly in this area.

### 2.2. Wind Erosion Models

#### 2.2.1. The NWESMC Model

The NWESMC model was developed based on the environment in northern China and wind tunnel experiments. This model was first applied to the evaluation of the Beijing-Tianjin Sand Source Control Engineering Project [30], then corrected according to the national large-scale survey data, and was improved and used in the first water conservancy survey [38]. The model establishes the empirical equations of arable land, grass land (forest land), and sand land in Equations (1)–(3), respectively. The actual soil wind erosion modulus of farmland [30] is as follows in Equation (1):(1)$$Q_{\mathrm{fa}}=10\times C\times (1-W)\times \sum_{j=1}\langle T_{j}\times \exp\{a_{1}+\frac{b_{1}}{z_{0}}+c_{1}\times \left[{\left({A\times U}_{j}\right)}^{0.5}\right]\}\rangle$$

The wind erosion modulus of grassland (forest land) is:(2)$$Q_{\mathrm{fgf}}=10\times C\times (1-W)\times \sum_{j=1}\left\{T_{j}\times \exp\left[a_{2}+b_{2}{\times \mathrm{VC}}^{2}+c_{2}/\left({A\times U}_{j}\right)\right]\right\}$$

The wind erosion modulus of sandy land is:(3)$$Q_{\mathrm{fs}}=10\times C\times (1-W)\times \sum_{j=1}\left\{T_{j}\times \exp\left[a_{3}+b_{3}{\times \mathrm{VC}+c}_{3}\times \ln\left({A\times U}_{j}\right)/\left({A\times U}_{j}\right)\right]\right\}$$
where Q_fa_ is the wind erosion of farmland (t hm^−2^ a^−1^), Q_fgf_ is the wind erosion of grassland (forest land) (t hm^−2^ a^−1^), and Q_fs_ is the wind erosion of sandy land (t hm^−2^ a^−1^). C is the scale revision factor (C = 0.0018), U_j_ is the j-th wind speed (m s^−1^) higher than the critical erosion wind speed in the hourly wind speed statistics of meteorological stations, T_j_ is the cumulative time (min) when the wind speed, U_j_, occurs in the month when the wind erosion activity occurs, VC is the vegetation coverage (%), Z_0_ is the surface aerodynamic roughness (cm), W is the topsoil humidity factor (%), A is the wind speed revision coefficient related to the underlying surface properties, a_1_, b_1_, and c_1_ are constant terms, with values of −9.208, 0.018, and 1.955, respectively, The values of a_2_, b_2_ and c_2_ are 2.4869, −0.0014, and −54.947, respectively. The values of a_3_, b_3_, and c_3_ are 6.1689, −0.0743, and −27.9613, respectively.

In NWESMC, potential wind erosion of half a month was computed, and the annual potential wind erosion was calculated by the half-monthly potential wind erosion.

#### 2.2.2. The RWEQ Model

RWEQ is one of the most commonly used empirical models for estimating wind erosion in farmland [39]. Because of the full consideration of climate and surface factors, data are easy to obtain and have been widely used; the RWEQ model has been applied to wind erosion areas in China, and the estimation results have been generally verified [40,41]. The basic governing equation of RWEQ is as follows in Equation (4) [39]:(4)$$Q_{x}{=Q}_{\max}\left[1-e^{{\left(\frac{x}{s}\right)}^{2}}\right]$$
where Q_x_ is the sediment flux at the block length × (distance from the upwind direction) (kg m^−1^), Q_max_ is the maximum sediment transport capacity of the wind force (kg m^−1^), and s is the key block length (m):(5)$$Q_{\max}=109.8(\mathrm{WF}\times \mathrm{EF}\times \mathrm{SCF}\times K^{\prime }\times \mathrm{COG})$$
(6)$$s=150.71{(\mathrm{WF}\times \mathrm{EF}\times \mathrm{SCF}\times K^{\prime }\times \mathrm{COG})}^{-0.3711}$$
where WF is a weather factor (kg m^−1^), EF is a soil erodible component (dimensionless), SCF is soil crust factor (dimensionless), K is the soil roughness factor (dimensionless), COG is the combined crop factor, including growing vegetation and withering vegetation (dimensionless).

The weather factor (WF) can be calculated using the following equation:(7)$$\mathrm{WF}=\overset{N}{\underset{i=1}{\sum }}\rho \frac{{{(U}_{2}-U_{t})}^{2}U_{2}}{\mathrm{gN}}\times N_{d}\times \mathrm{SW}\times \mathrm{SD}$$
where U_2_ is the wind speed (m s^−1^) at a height of 2 m, which can be converted from wind speed observed at standard anemometer heights using the 1/7 power expression method [11]. Ut is the threshold wind speed (m s^−1^) at a height of 2 m; Guo suggested that the threshold wind speed for arable land in north-central parts of China is 5 m s^−1^ [42]. N is the observation times of wind speed, N_d_ is the number of days in the period (usually 15 days), g is gravitational acceleration (m s^−2^), SW is a soil wetness factor (dimensionless), and SD is a snow cover factor (dimensionless).

The soil erodibility factor (EF) and the soil crust factor (SCF) can be calculated as follows:(8)$$\mathrm{EF}=\frac{29.9+0.31\mathrm{Sa}+0.17\mathrm{Si}+0.+33\mathrm{Sa}/\mathrm{Cl}-2.59\mathrm{OM}-{0.95\mathrm{CaCO}}_{3}}{100}$$
(9)$$\mathrm{SCF}=\frac{1}{1+{0.0066(\mathrm{Cl})}^{2}+{0.021(\mathrm{OM})}^{2}}$$
where Sa is the soil sand content (%), Si is the soil silt content (%), Sa/Cl is the ratio of soil sand and clay content, OM is the organic matter content (%), and CaCO_3_ is the calcium carbonate content (%). Because of the small interannual variation of soil texture and organic matter content, it is assumed that the soil erodibility and crust factors will not change with time.

The combined crop factor (COG) is determined by the flat residues (SLR_f_), standing residues (SLR_s_) and crop canopy (SLR_c_) factors:(10)$${\mathrm{COG}=\mathrm{SLR}}_{f}\times {\mathrm{SLR}}_{s}\times {\mathrm{SLR}}_{c}$$

In the APEC region, crop residues are generally used for heating and fuel and most cropland contains no crop residue [32]. Therefore, the wind erosion is governed by vegetation cover, and the COG is determined by SLR_c_ [27,32,43]. SLR_c_ is calculated as in Equation (11):(11)$${\mathrm{SLR}}_{c}{=e}^{-{5.614(\mathrm{cc}}^{0.7366})}$$
where cc is the fraction of soil surface covered with crop canopy. The cc is obtained from the normalized difference vegetation index (NDVI) [44]:(12)$$\mathrm{cc}=\frac{{\mathrm{NDVI}-\mathrm{NDVI}}_{s}}{{\mathrm{NDVI}}_{v}{-\mathrm{NDVI}}_{s}}$$
where NDVI is the normalized difference vegetation index, NDVI_s_ is the value of bare soil pixels, and NDVI_v_ is the value of vegetated pixels.

In RWEQ, the potential wind erosion of half a month was calculated, and the annual potential wind erosion was determined from the half-monthly potential wind erosion.

#### 2.2.3. The WEPS Model

WEPS—a physics-based model—can simulate weather, surface conditions, field management, and wind erosion in time steps less than daily (e.g., hourly) [45,46]. The WEPS manual stipulates that, when the maximum wind speed at a height of 10 m exceeds 8 m s^−1^, the wind erosion submodule will run [47]. The WEPS has been successfully extended to non-agricultural disturbed lands for simulating regional potential wind erosion in Western U.S. and Northern China [48,49,50,51]. The calculation of each step and the required basic equations are as follows:(13)$$Q={0.4\times u}_{*}{}^{2}{(u}_{*}-{0.8\times u}_{*t})$$
where Q is the emission transport capacity (kg m^−1^ s^−1^), u_*_ is the friction velocity (m s^−1^), and u_*t_ is the static threshold friction velocity (m s^−1^). It reveals that the driving force of sand transport is when the friction velocity of u_*_ is greater than the static threshold friction velocity of u_*t_.

Friction velocity at the sub-region is calculated in two steps. First, the friction velocity at the weather station, where wind speeds are measured, is calculated using the log-law profile:
(14)$$u_{*f}=\frac{0.4\times u}{\ln\left(\frac{z}{z_{0f}}\right)}$$
where u_*f_ is friction velocity at the weather station (m s^−1^), u is wind speed at the weather station (m s^−1^), z is anemometer height at the weather station (mm) (wind speeds were adjusted to 10 m height in the WEPS database), z_0f_ is aerodynamic roughness at the weather station, which is assumed to be 25 mm in WEPS.

Second, when there is no vegetation, the calculation method of friction velocity is as follows:(15)$$u_{*}{=u}_{*f}\times {\left(\frac{z_{0}}{z_{0f}}\right)}^{0.067}$$
where z_0_ is the local aerodynamic roughness as in [38].

The static threshold friction velocity takes into account the surface soil texture, flat biomass and surface wetness, and the calculation formula is as follows:(16)$$u_{*t}={\mathrm{WUB}}_{*\mathrm{ts}}+{\mathrm{WUC}}_{*\mathrm{ts}}+{\mathrm{WUCW}}_{*\mathrm{ts}}$$
where WUB_*ts_ is the static threshold friction velocity of bare surface (m s^−1^); the minimum static threshold friction velocity for field surfaces was generally set to be 0.35 m s^−1^ [47]; WUC_*ts_ is the change in static threshold friction velocity caused by flat biomass cover (m s^−1^), and WUCW_*ts_ is the increase in static threshold friction velocity from surface wetness (m s^−1^).

Because of the large area of APEC, some soil texture data are difficult to obtain, resulting in a large error of WUB_*ts_. Previous studies [52] replaced u_*t_ in the SWEEP model with u_*t_ as in Lu and Shao [53] and have been well verified. Therefore, this study replaced WUB_*ts_ with u_*t_ on smooth and dry surfaces in the IWEMS model. The formula for WUB_*ts_ is as follows:(17)$${\mathrm{WUB}}_{*\mathrm{ts}}=\sqrt{\beta_{1}\times \left(\sigma_{p}\mathrm{gd}+\frac{\beta_{2}}{\rho d}\right)}$$
where δ_p_ is the particle-to-air density ratio, d is the particle diameter (m), g is the acceleration of gravity, 9.8 m s^−2^, ρ is the air density (kg m^−3^), β_1_ is 0.0123, and β_2_ is 3 × 10^−4^ kg s^−2^.

The formula for WUC_*ts_ is as follows:(18)$${\mathrm{WUC}}_{*\mathrm{ts}}={0.02+\mathrm{SFC}}_{\mathrm{cv}}$$
(19)$${\mathrm{SFC}}_{\mathrm{cv}}=\left({-\mathrm{SF}}_{\mathrm{cv}}\right){\times \mathrm{BFF}}_{\mathrm{cv}}$$
where SFC_cv_ is the fraction change in soil surface area protected from emission, SF_cv_ is the soil fraction covered by clod/crust and rock so it does not emit, and BFF_cv_ is the biomass fraction of flat cover.

The formula for WUCW_*ts_ is as follows:(20)$${\mathrm{WUCW}}_{*\mathrm{ts}}=0.48\times \frac{{\mathrm{HRO}}_{\mathrm{WC}}}{{\mathrm{HR}15}_{\mathrm{WC}}},\;\frac{{\mathrm{HRO}}_{\mathrm{WC}}}{{\mathrm{HR}15}_{\mathrm{WC}}}>0.2$$
where HROwc is the surface soil water content (kg kg^−1^), and HR15wc is the surface soil water content at 1.5 MPa (kg kg^−1^).

In WEPS, the daily potential wind erosion was evaluated, and the annual potential wind erosion was calculated from the daily potential wind erosion.

#### 2.2.4. The IWEMS Model

IWEMS is a model developed based on arid and semi-arid regions in Australia to predict wind erosion processes at regional and national scales [54]. Recently, the IWEMS were widely used to evaluate regional wind erosion and dust emission across the northern China [33,55,56]. The streamwise saltation flux Q(ds) (kg m^−1^ s^−1^) for soil of uniform particle size ds can be estimated using Owen’s model [57]:(21)$$Q\left(d_{s}\right)=\{\begin{array}{c} \frac{c_{o}A_{c}\rho_{a}u_{*}{}^{3}}{g}\left[1-{\left(\frac{u_{*t}}{u_{*}}\right)}^{2}\right]{,\&u}_{*}\geq u_{*t}\\ {0,u}_{*}{<u}_{*t} \end{array}$$
where A_e_ is the fraction of erodible area and Co is the Owen coefficient. In theory, Co is not a constant but dependent on ω_t_(ds)/u_*_, equal to 0.25 + ω_t_(ds)/3u_*_ in Owen’s original formulation. The typical value of Co is around one, but with a considerable scatter.

Threshold friction velocity is estimated by:(22)$$u_{*t}{(d}_{s}{;\lambda ;\theta )=u}_{*t}{(d}_{s}{)f}_{\lambda }{(\lambda )f}_{\omega }\left(\theta \right)$$
where u_*t_(ds,λ,θ) is the threshold friction velocity of sand particles with diameter ds in the presence of vegetation and soil moisture (m s^−1^), λ is the frontal area of the roughness element (m^2^), f_λ_(λ) is a function that modifies the threshold friction velocity to reflect the roughness elements, θ is the volumetric soil moisture (m^3^ m^−3^), f_w_(θ) is a function that corrects threshold friction velocity for soil moisture, and u_*t_(ds) is the threshold friction velocity under the ideal condition that the surface is covered by loose sand particles of uniform and spherical shape. The threshold friction velocity under ideal conditions, u_*t_(ds) can be expressed by an equation proposed by Shao [54]:(23)$$u_{*t}{(d}_{s})=\sqrt{\beta_{1}\left(\sigma_{p}{\mathrm{gd}}_{s}+\frac{\beta_{2}}{{\rho d}_{s}}\right)}$$
where δ_p_ is the particle-to-air density ratio, ds is the particle diameter (m), g is the acceleration of gravity, 9.8 m s^−2,^ ρ is the air density (kg m^−3^), β_1_ is 0.0123 and β_2_ is 3 × 10^−4^ kg s^−2^.

The formula for f_λ_(λ) is as follows:(24)$$f_{\lambda }(\lambda )=\frac{u_{*t}{(d}_{s},\lambda )}{u_{*t}{(d}_{s})}={\left({1-m}_{r}\sigma_{r}\lambda \right)}^{1/2}{\left({1+m}_{r}\sigma_{r}\lambda \right)}^{1/2}$$
where m_r_ is a tuning parameter with a value less than one, which accounts for non-uniformities in the surface stress distribution, δ_r_ is the ratio of basal to frontal area (δ_r_ = η/λ) of the roughness elements, and β_r_ = C_p_/C_s_ is the ratio of the pressure-drag coefficient to the friction-drag coefficient.

The formula for f_w_ is as follows:(25)$$f_{w}={\left[1+A{\left({\theta -\theta }_{r}\right)}^{b}\right]}^{1/2}$$
where θ_r_ is air-dry soil moisture (m^3^ m^−3^), and A and b are dimensionless parameters.

In IWEMS, daily potential wind erosion was evaluated, and the annual potential wind erosion was determined from the daily potential wind erosion.

### 2.3. Data Preparation

The meteorological data used in this research were obtained from the national station data from 2000 to 2012 provided by the China Meteorological Data Service Center (CMDC) (http://data.cma.cn, accessed on 10 July 2020), of which the meteorological data mainly include wind speed, wind direction, temperature, precipitation, and sunshine hours. NDVI data were obtained from MODIS data products provided by USGS website (https://www.usgs.gov, accessed on 12 July 2020), of which MOD13A2, MOD11A2, and MOD09A1 were used. The soil properties data were mainly determined from the Chinese soil data set of the World Soil Database (HWSD) provided by the Cold and Arid Regions Scientific Data Center (http://westdc.westgis.ac.cn, accessed on 12 July 2020). Digital elevation data used the China 1-km resolution digital elevation model data set provided by the Cold and Arid Regions Scientific Data Center (http://westdc.westgis.ac.cn, accessed on 12 July 2020) Table 1. The land use data of 1-km resolution for 2000, 2005, and 2010 were provided by the Resources and Environment Data Cloud Platform (http://www.resdc.cn, accessed on 15 July 2020). Farmland, grassland, and sands (or desert) were selected to estimate potential wind erosion. Furthermore, we estimated potential wind erosion for a square 1 ha (100 m × 100 m) field [32]. Aerosol optical thickness (AOD) data are provided by the TGP group, Institute of Remote Sensing and Digital Earth, Chinese Academy of Sciences (http://www.tgp.ac.cn/, accessed on 15 July 2020) [58]. In addition, the flow chart of this study is presented in Figure 2. In this research, the “Classification Standard of Wind Erosion (SL190-2007)” was used to classify the potential wind erosion hazard (weak, slight, moderate, severe, very severe, or catastrophic) [59].

## 3. Results

### 3.1. Potential Wind Erosion for Different Hazards

The potential wind erosion from 2000 to 2012 for the four models with different erosion hazards is shown in Table 2. In NWESMC, more than 80% of the land suffered weak or slight wind erosion in the region, and no catastrophic wind erosion occurred. When compared with NWESMC, the area of “slight” hazard class increased significantly, but the “weak” hazard agricultural decreased considerably in RWEQUATION The percentages of potential wind erosion hazards were similar for WEPS and IWEMS, with about 60% of the land undergoing weak or slight wind erosion and more than 20% of the land suffering very severe and catastrophic wind erosion. In summary, IWEMS generally yielded the highest potential erosion, while NWESMC estimated the lowest potential wind erosion. The average potential wind erosion was 12.58, 25.87, 52.63, and 58.72 t hm^−2^ a^−1^ for NWESMC, RWEQ, WEPS, and IWEMS, respectively.

### 3.2. Spatial Variation of Potential Wind Erosion

The geographic distributions of the potential wind erosion were somewhat similar, but the magnitudes of potential wind erosion were different for different models (Figure 3). For NWESMC, very severe wind erosion scattered in the Horqin Sands and the Hunshan Dake Sands with a sporadic distribution pattern. Moderate wind erosion occurred in the Mu Us Sands. For RWEQ, the highest potential erosion with catastrophic hazard occurred in the southeast of the APEC, near and within the Horqin Sands. The other high erosion region was mainly scattered in the Hunshan Dake Sands. The regional patterns of wind erosion hazard for WEPS and IWEMS were markedly different from those of NWESMC and RWEQ, the catastrophic and very severe hazard wind erosion regions appeared in the Hunshan Dake Sands and Mu Us Sands, while the Horqin Sands still suffered catastrophic wind erosion.

### 3.3. Temporal Distribution of Potential Wind Erosion

Although the potential wind erosion showed obvious annual fluctuation, it decreased significantly from 2000 to 2012 (Figure 4). The ANOVA test (2-tailed) by SPSS 23.0 software (Armonk, NY, USA) was used to estimate the trend in the average potential wind erosion for different models, and the trend of change was roughly the same and presented a significant linear decline from 2000 to 2012 (*p* < 0.01). The average potential wind erosion reached the highest value in 2001, with values of 18.82, 54.88, 128.44, and 104.13 t hm^−2^ a^−1^ for NWESMC, RWEQ, WEPS, and IWEMS, respectively. The average potential wind erosion was lowest in 2011, with values of 8.60, 9.18, 13.33, and 31.25 t hm^−2^ a^−1^ for NWESMC, RWEQ, WEPS, and IWEMS, respectively.

### 3.4. Potential Wind Erosion under Different Land Use

Most territory of the APEC was covered by farmland, grassland, and sand land [32]. The average predicted potential wind erosion under different land use varied remarkably (Figure 5). The average potential wind erosion values of grassland, farmland, and sand land calculated by NWESMC were 1.37, 15.85, and 34.3 t hm^−2^ a^−1^, respectively. The average potential wind erosion values of grassland, farmland, and sand land calculated by RWEQ, WEPS, and IWEMS showed similar successive increases. The ratios between average potential wind erosion of grassland and sand land for NWESMC, WEPS, IWEMS, and RWEQ were 0.04, 0.12, 0.15, and 0.21, respectively.

## 4. Discussion

### 4.1. Models’ Verification and Applicability

Quantitative regional measured values of soil loss by wind erosion are not available for the APEC. However, some studies describing field soil loss by wind erosion were performed using different methods. To evaluate the performance of the four wind erosion models, 31 observed wind erosion datasets obtained by different observation methods were collected from published literature (Table 3). The 31 locations of these observed data scatter in the main land use of the APEC (Figure 6). To investigate the relationship between observed wind erosion data and values obtained using the four models, linear regression analysis and Sutcliffe efficiency coefficient (NSC) analysis were conducted (Figure 7). The coefficients of determination (R^2^), NSC, and the magnitudes of the slopes obtained through the SPSS and used to evaluate how well these models predict wind erosion [60,61]. The R^2^ values suggest that predicted wind erosion was linearly related to measured wind erosion. The magnitudes of the slopes indicate that predicted wind erosion was generally smaller than observed wind erosion (Figure 7). The values of the R^2^ (*p* < 0.05) and NSC demonstrated that the performance of RWEQ, WEPS, and IWEMS were relatively satisfactory, while performance of the NWESMC was relatively poor. Furthermore, the annual average potential wind erosions are 12.82, 26.97, 54.23, and 61.14 t hm^−2^ a^−1^ for NWESMC, RWEQ, WEPS, and IWEMS, which are similar to or slightly higher than the previous studies [62,63]. Figure 8 presented the spatial distribution and interannual variation the MODIS satellite AOD data from 2003 to 2010. High AOD generally scattered in the Mu Us Sands, Horqin Sands, and the Hunshan Dake Sands, which were similar to the spatial distribution of severe wind erosion (Figure 3). In addition, the interannual variations of AOD is also consistent with the trends of annual potential wind erosion obtained from the four models (Figure 4).

For the model structure, the NWESMC and RWEQ belonged to the empirical model, and the WEPS and IWEMS were mechanistic models [34]. These models were widely used for regional wind erosion evaluation in the arid and semi-arid China to date [64]. In this study, the version of the NWESMC was improved according to the national large-scale survey data in the first soil and water conservation survey [38]. Abundant studies demonstrated that the RWEQ is capable of modeling daily, monthly, and annual (potential) wind erosion across field, regional, and global scales after some adjusting [32,64,65]. Here, we used the up-scaling method of RWEQ proposed by Guo [32]. Determining the friction threshold wind velocity (u_*t_) is the key step when using the WEPS [48,49,51]. The regional version of the WEPS was used to calculate daily u_*t_ and further model potential wind erosion in the APEC. The IWEMS can incorporate geographic information systems and remote sensing data to estimate regional wind erosion. Du adjusted the IWEMS for extending it to northern China based on observed wind erosion data [55]. It revealed that the IWEMS can predict regional wind erosion and dust emission. The revised IWEMS was used to model potential wind erosion in the APEC. The above analysis indicated that the revised or improved models could evaluate the temporal trends and spatial patterns of potential wind erosion in the APEC.

### 4.2. Factors Impacting on Regional Potential Wind Erosion Modeling

Wind erosivity (e.g., wind speed, turbulence), soil erodibility (e.g., soil aggregate size distribution, crust, moisture), and surface coverage and roughness (e.g., canopy or residue coverage, microrelief) govern the wind erosion process [13,15,80,81,82,83]. At a regional scale, wind erosion modeling is generally influenced by remote sensing vegetation coverage, soil moisture, and upscaled meteorological data [1,15]. We further explore how these factors influence regional potential wind erosion evaluation.

Only small areas showed significant correlation between potential wind erosion and annual average precipitation or temperature (Table 4). However, the areas with significant positive correlation (*R* > 0.74, *p* < 0.05) between potential wind erosion and annual average wind speed were 40.68%, 42.37%, 27.01%, and 31.53% of the total territory of the APEC for NWESMC, RWEQ, WEPS, and IWEMS, respectively. The regions with significant negative correlation (*R* < −0.74, *p* < 0.05) between potential wind erosion and annual average soil moisture or vegetation coverage were smaller than those with a significant positive correlation but showed similar trends. For NWESMC, the areas with significant positive (negative) correlation between potential wind erosion and wind speed, soil moisture and vegetation coverage decreased, with the same trend observed for RWEQ, WEPS, and IWEMS. These results indicated that the magnitudes of sensitivity of wind speed, soil moisture, and vegetation coverage to regional wind erosion modeling successively reduced.

Wind speed is generally considered as the primary driving factor initiating wind erosion while high vegetation coverage and wet surface soil can significantly lower soil susceptibility to wind erosion [43,84]. The sensitivity of parameters of WEPS, NWESMC, RWEQ, and IWEMS were determined by the previous studies about [30,84,85]. These results concluded that wind speed was the most sensitive input parameter and vegetation and soil moisture could curb the wind erosion. Vegetation with different land-use or soil moisture was also important for WEPS regional wind erosion modeling [51]. Shao revealed that wind speed (friction wind velocity) and vegetation coverage soil moisture were very sensitive factors when modeling wind erosion with IWEMS [12]. Studies of regional wind erosion modeling by RWEQ indicated that wind speed and the soil crust factor (which occurred after rain) were very sensitive inputs [32,43]. NWESMC used the cumulative time of erosive wind speeds and vegetation coverage to directly calculate soil loss by wind erosion [30]. Consequently, wind speed, vegetation coverage, and soil moisture can be summarized as the main factors affecting simulated regional potential wind erosion.

### 4.3. Limitations and Future Perspectives

In fact, the wind erosion predicted by the models was the wind erosion potential or not the real on-site wind erosion in this study. More detail model calibration may be necessary based on a long-term observed wind erosion dataset to obtain more reliable large-scale wind erosion. These wind erosion models with diverse computation structure originated from different countries based on various geographic conditions. The selection of a particular model is generally dependent on the available databases and the specific requirements [15]. In practice, these models have been widely used in regional wind erosion assessment. Some studies calibrated [55,86] or validated [44,55,87] the models for regional wind erosion estimation. However, other studies directly evaluated regional wind erosion without calibration and validation of wind erosion models [26,27,32,48,64,88,89]. Here, we demonstrated that various values of potential wind erosion were yielded by different models using the same available databases. Accordingly, it is necessary to systematically calibrate and validate the selected wind erosion model before extending it to regions with differing geographic conditions.

Theoretically, the reliable calibration and validation of a wind erosion model requires a long-term observed wind erosion dataset with a consistent measurement method. In water erosion research, experimental plots have been the standard method since the 1930s [35]. In contrast, aeolian sand samplers have been deployed and extensively used for various different purposes. For example, Chepil generally used a rectangle field to monitor sand transport [90]. Wind erosion circles were popular when developing RWEQ and WEPS [11,45]. Webb proposed a national wind erosion research network based on square experimental plots (100 m × 100 m) [1]. Some studies used sand traps with a uniform grid network [65,91]. The above analysis indicates that a long-term standard wind erosion monitoring network is urgently required. The standard network requirements would include the shape of experimental plots, meteorological data observation procedure, field soil sampling method, sand trap type, and its deployment scheme [13,34,92].

## 5. Conclusions

In this study, NWESMC, RWEQ, WEPS, and IWEMS were used to simulate the temporal and spatial pattern of the potential wind erosion in the APEC from 2000 to 2012. The impacts of meteorological data, soil moisture, and vegetation coverage on potential wind erosion were discriminated and quantified. The main conclusions were:The potential wind erosion values predicted by the four models were correlated with the observed wind erosion collected from published documents, but the correlation coefficients between the predicted and the measured wind erosion data for the four models vary greatly;The values of average potential wind erosion were different while the spatial pattern of potential wind erosion was similar for different wind erosion models. Most areas of APEC suffered from weak and slight hazards of wind erosion, while severe and catastrophic hazards of wind erosion mainly occurred in the Horqin, Mu Us, and Hunshan Dake sands;The temporal trends of annual potential wind erosion were similar and the total potential wind erosion decreased significantly from 2000 to 2012;The average potential wind erosion of grassland, farmland, and sand land calculated by NWESMC, RWEQ, WEPS, and IWEMS showed similar successive increases.Wind speed, soil moisture, and vegetation coverage were the dominant factors affecting regional wind erosion estimation.

These results further revealed that it is necessary to comprehensively calibrate and validate the selected wind erosion models. A long-term standard wind erosion monitoring network is urgently required.

## Figures and Tables

**Figure 1 ijerph-19-09538-f001:**
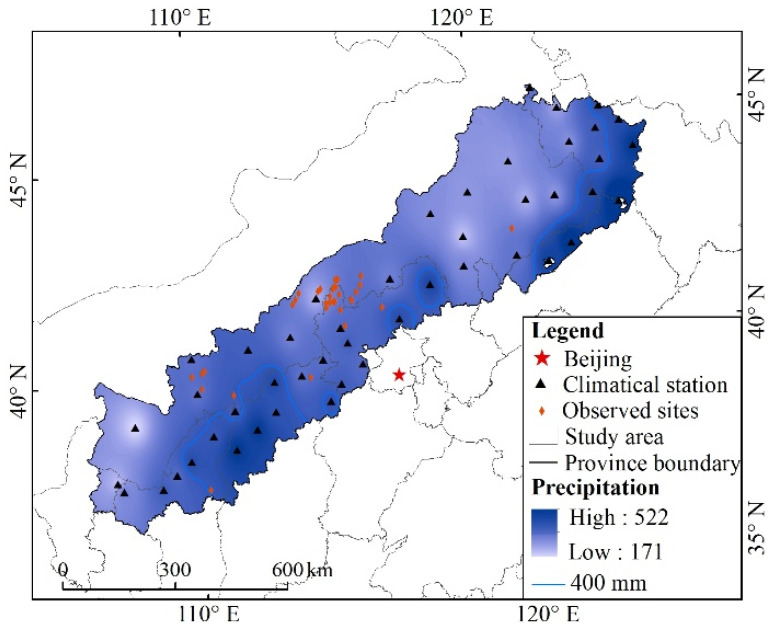
Schematic map of the Agro-Pastoral Ecotone of northern China (APEC).

**Figure 2 ijerph-19-09538-f002:**
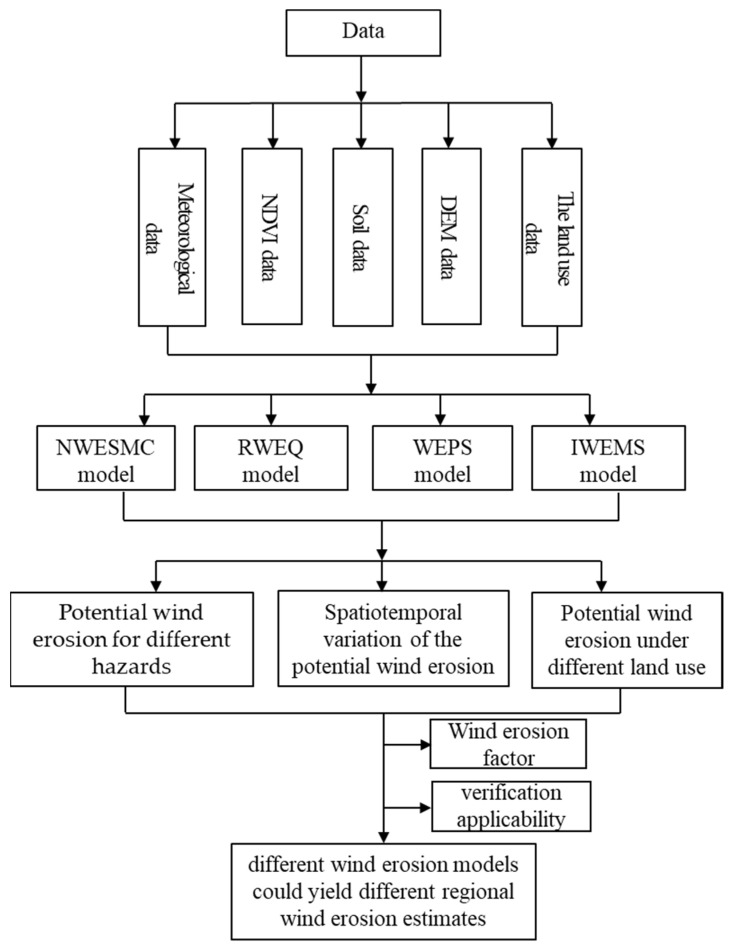
Flow chart of the study.

**Figure 3 ijerph-19-09538-f003:**
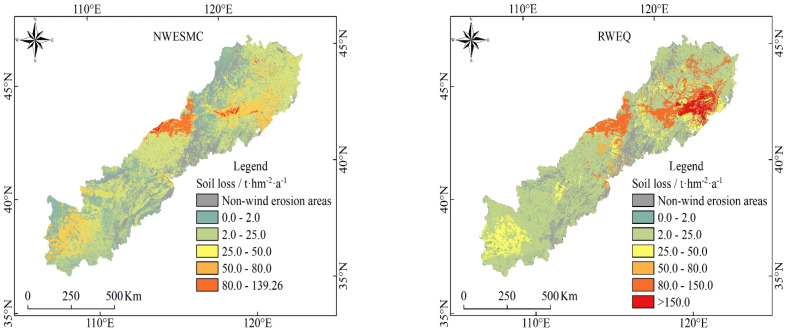
Spatial distribution of potential wind erosion for the NWESMC, RWEQ, WEPS, and IWEMS in the APEC.

**Figure 4 ijerph-19-09538-f004:**
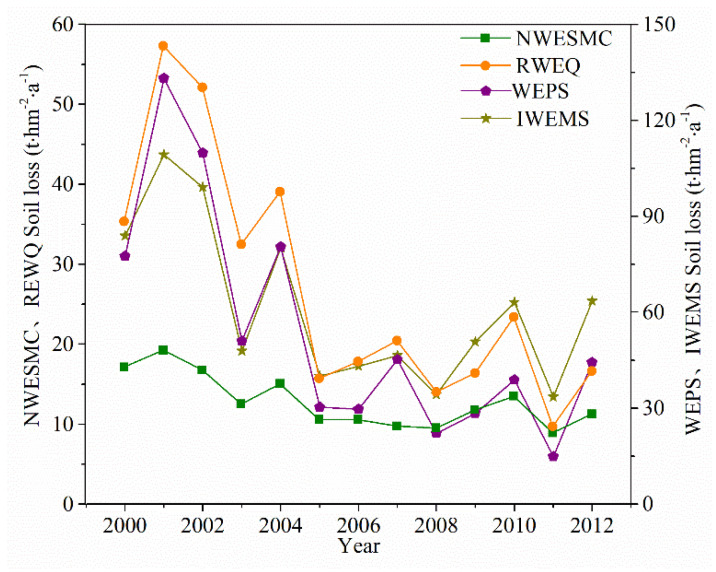
Interannual variation of average potential wind erosion of different models in the agro-pastoral ecotone of northern China (APEC) from 2000 to 2012.

**Figure 5 ijerph-19-09538-f005:**
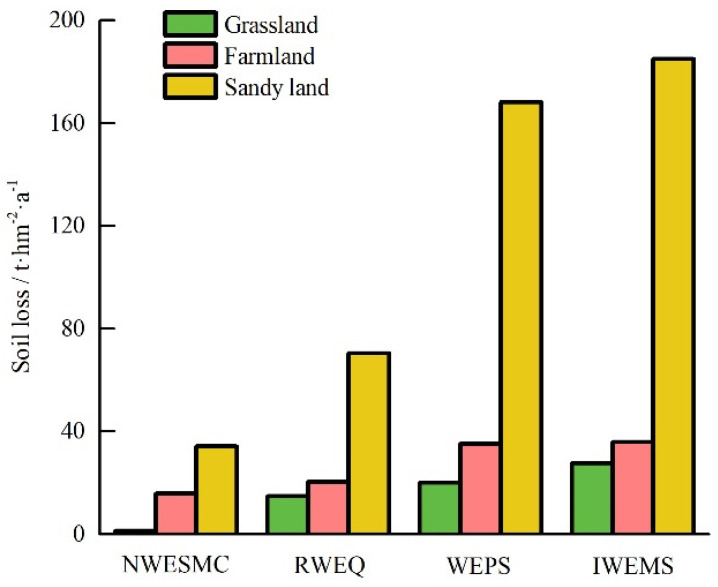
Potential wind erosion under different land use for different wind erosion models.

**Figure 6 ijerph-19-09538-f006:**
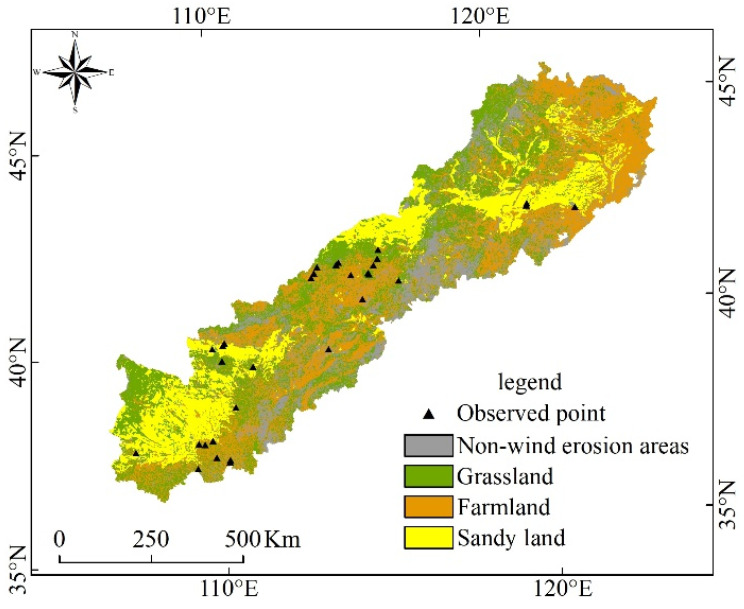
Distribution of the observed wind erosion sites.

**Figure 7 ijerph-19-09538-f007:**
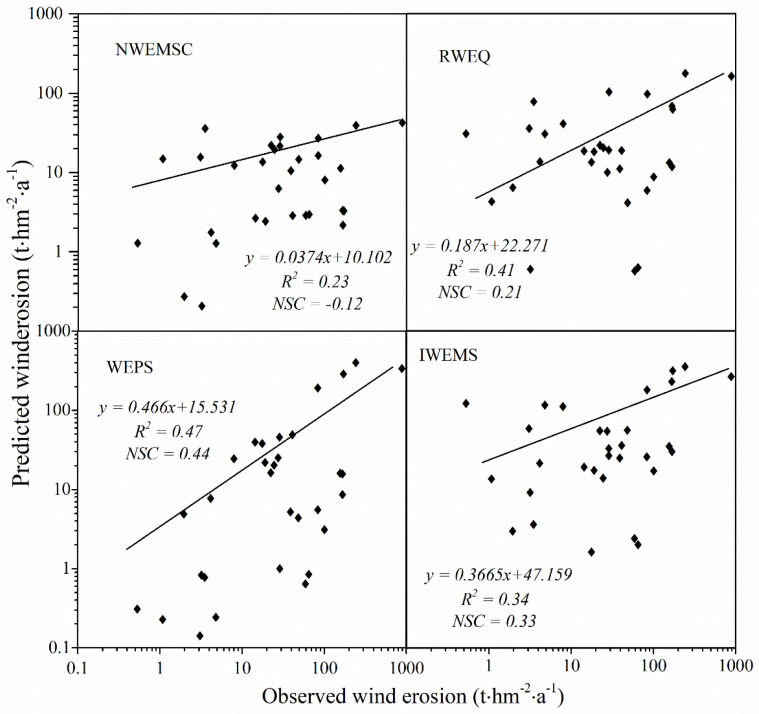
The relationship between the wind erosion predicted by the NWESMC, RWEQ, WEPS, and IWEMS models and that retrieved from published literature (Table 3).

**Figure 8 ijerph-19-09538-f008:**
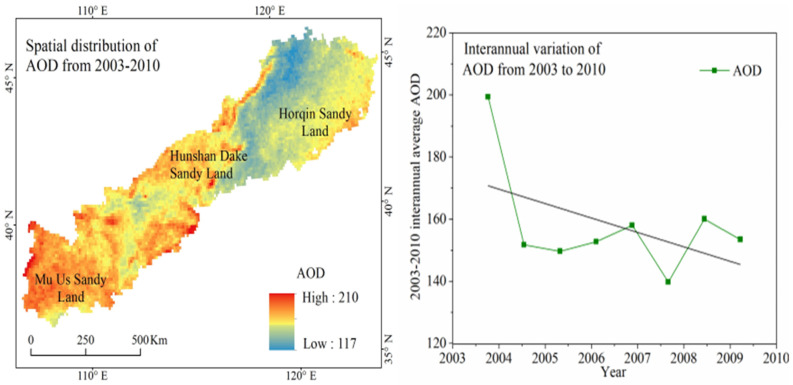
Spatial distribution and interannual variation of Aerosol Optical Depth (AOD) from 2003 to 2010 in the agro-pastoral ecotone of northern China (APEC). The black line indicates the regression-trend line for the AOD from 2003 to 2010. The data set is provided by the TGP group, Institute of Remote Sensing and Digital Earth, Chinese Academy of Sciences (http://www.tgp.ac.cn/, accessed on 10 July 2020).

**Table 1 ijerph-19-09538-t001:** Data requirements for wind erosion modeling.

Data Types	Temporal Resolution	Spatial Resolution	Format	Web Sites
Meteorological data	Hourly/Daily	N/A	Text	http://data.cma.cn accessed on 10 July 2020
Normalized difference vegetation index (NDVI)	16 days	1 km	Raster	https://www.usgs.gov accessed on 12 July 2020
Soil data	N/A	1 km	Raster	http://westdc.westgis.ac.cn accessed on 12 July 2020
Digital Elevation Model (DEM) data	N/A	1 km	Raster	http://westdc.westgis.ac.cn accessed on 12 July 2020
The land use data	Annual	1 km	Raster	http://www.resdc.cn accessed on 15 July 2020
Aerosol optical depth (AOD)	Annual	0.1°	Raster	https://data.tpdc.ac.cn accessed on 15 July 2020

Note: In the meteorological data, the wind speed data are hourly data, and the other required meteorological data are daily data. N/A indicates that the information is not available.

**Table 2 ijerph-19-09538-t002:** Potential wind erosion hazards for different wind erosion models.

Class/Range (t hm^−2^ a^−1^)	NWESMC	RWEQ	WEPS	IWEMS
	Area of the Class (km^2^)/Percent of Total Area for the Class (%)			
Weak/0–2	222 298/40.88	88 186/16.23	174 601/32.12	138 243/25.43
Slight/2–25	240 184/44.17	325 643/59.94	174 257/32.06	182 649/33.60
Moderate/25–50	61 382/11.29	54 389/10.01	72 945/13.42	82 648/15.20
Severe/50–80	18 178/3.34	13 818/2.54	34 763/6.40	31 292/5.76
Very Severe/80–150	1 747/0.32	48 251/8.88	27 484/5.06	36 604/6.73
Catastrophic/>150	0/0.0	13 015/2.40	59 474/10.94	72 133/13.27

Note: National Wind Erosion Survey Model of China (NWESMC), Revised Wind Erosion Equation (RWEQ), Wind Erosion Prediction System (WEPS), Integrated Wind Erosion Modeling System (IWEMS).

**Table 3 ijerph-19-09538-t003:** Observed wind erosion collected from published documents.

Site No.	Land Use	Method	Wind Erosion (t hm^−2^ a^−1^)	Reference
1	Sand	Field Survey	243.00	Zhao et al., 1988 [66]
2	Farmland	Sand trap	883.30	Xu et al., 1993 [67]
3	Farmland	Particle-size distribution comparison method	14.40	Dong et al., 1997 [68]
4	Farmland		24.60	
5	Farmland		19.05	
6	Farmland		41.10	
7	Farmland		28.80	
8	Sand	Sand trap	83.95	Li et al., 2005 [69]
9	Farmland	^137^Cs	28.97	Zhao et al., 2005 [70]
10	—	Sediment analysis	172.23	Shi et al., 2006 [71]
11			8.02	
12			156.57	
13			167.43	
14			167.97	
15			22.46	
16			39.08	
17	Farmland	Sand trap	1.08	Wang et al., 2006 [72]
18	Grassland	^137^Cs	3.51	Liu et al., 2007 [73]
19	Grassland		4.18	
20	Grassland		0.53	
21	Grassland		4.80	
22	Grassland		3.10	
23	—	Sediment analysis	101.00	Li et al., 2011 [74]
24	Farmland	Field Survey	27.50	Guo et al., 2016 [75]
25	Farmland	^137^Cs	17.65	Jiang, 2010 [76]
26	Farmland	^137^Cs	83.62	Zhang et al., 2010 [77]
27	Farmland	^137^Cs	59.00	Li et al., 2016 [78]
28	Grassland		3.20	
29	Farmland		65.00	
30	Sand		48.50	
31	Farmland	Sand trap	1.96	Guo et al., 2019 [79]

**Table 4 ijerph-19-09538-t004:** Spatial correlation analysis between potential wind erosion and wind speed, soil moisture, vegetation, precipitation, and temperature.

Model	Correlation Analysis	Wind Speed (M S^−1^)	Soil Moisture (%)	Vegetation (%)	Precipitation (mm)	Temperature (°C)
		Area of the Correlation Level (km^2^)/Percent of Total Area for the Correlation Level (%)				
NWESMC	Significant negative correlation	1 325/0.31	134 697/31.21	80 325/18.64	6 010/1.39	12 011/2.78
	Negative correlation	20 734/4.79	249 886/57.89	200 194/46.46	363 203/83.98	71 297/16.48
	No correlation	2/0.00	3/0.00	14/0.00	1/0.00	9/0.00
	Positive correlation	234 479/54.21	44 487/10.31	137 871/32	63 020/14.57	315 922/73.04
	Significant positive correlation	175 967/40.68	2 567/0.59	12 486/2.9	273/0.06	33 268/7.69
RWEQ	Significant negative correlation	4 027/0.93	94 338/21.86	53 579/12.43	11 852/2.74	11 144/2.58
	Negative correlation	34 018/7.87	315 636/73.13	193 589/44.93	394 135/91.13	46 389/10.73
	No correlation	0/0.00	0/0.00	6/0.00	1/0.00	30/0.01
	Positive correlation	210 974/48.78	21 680/5.02	168 168/39.03	26 280/6.08	366 320/84.70
	Significant positive correlation	183 260/42.37	285/0.07	15 964/3.71	11/0.00	8 396/1.94
WEPS	Significant negative correlation	511/0.12	105 471/24.44	47 839/11.1	5 184/1.2	19 114/4.42
	Negative correlation	48 425/11.2	283 505/65.68	239 370/55.55	340 884/78.81	68 562/15.85
	No correlation	2/0.00	11/0.00	2/0.00	1/0.00	32/0.01
	Positive correlation	266 665/61.65	40 761/9.44	141 026/32.73	83 639/19.34	338 221/78.2
	Significant positive correlation	116 841/27.01	2 245/0.52	3 067/0.71	2 736/0.63	6 515/1.51
IWEMS	Significant negative correlation	101/0.02	105 396/24.42	94 572/21.95	11 975/2.77	29 598/6.84
	Negative correlation	24 081/5.57	243 576/56.43	243 201/56.44	306 282/70.81	128 717/29.76
	No correlation	0/0.00	8/0.00	12/0.00	0/0.00	31/0.01
	Positive correlation	271 953/62.88	76 719/17.77	90 823/21.08	110 984/25.66	243 031/56.19
	Significant positive correlation	136 377/31.53	5 928/1.37	2 267/0.53	3 271/0.76	31 135/7.20

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
