# Peer review of "Regional Potential Wind Erosion Simulation Using Different Models in the Agro-Pastoral Ecotone of Northern China"

_ijerph, 2022, doi:10.3390/ijerph19159538_

Round 1

Reviewer 1 Report

Comments on “Regional potential wind erosion simulation…”

This study compared the simulation results of region potential wind erosion in Northeast China among four wind erosion models and further evaluated model results using limited observations. The Method section gave a detailed description of the implementation of the four models, which can benefit the wind erosion modelers in terms of getting better understanding of the differences among the four models. The results of this study are supported by their analysis and very interesting. My main suggestion is performing a deeper analysis on the decreasing trend in the potential wind erosion as shown in Figure 3. Two things can be done regarding this suggestion: 1) using observations or satellite retrievals to cross-validate the simulated decreasing trend and 2) figuring out what factors drive the decreasing trend. This would be one of major findings of this study. One more suggestion is to compare the models’ sensitivities of the potential wind erosion to different factors, such as wind, land type, and vegetation coverage. Overall, I think this study could be an important contribution to the wind erosion community.

Comments:

1.     In Abstract the four acronyms should be defined in their first appearance.

2.     Line 35: should be dust emissions.

3.     Line 80–81: There’s no need to mention the names of these provinces since they are not marked in Figure 1.

4.     Figure 1: Please use precipitation map to replace the topography map, because the APEC is defined closely based on precipitation rather than topography. Authors may add another panel in Figure 1 to include land use types.

5.     Figure 3: Please use lines of different colors to represent different model results.

6.     Trend in Section 3.3: How the trends are estimated? Do they pass the statistically significant test?

7.     The simulated decreasing trend in potential wind erosion should be cross-validated using observation and/or satellite data. For example, PM10 data and satellite retrieved AOD can be used. What factors drive the decreasing trend should be also investigated, such as winds, land use type, precipitation, vegetation, and so on?

8.     Discuss the potential role of irrigation in the wind erosion in this region.

9.     Figure 5: Most of the observational data points are located near zero, so I suggest use the log-scale for both x- and y-axis to zoom in the area where data density is high.

Author Response

This study compared the simulation results of region potential wind erosion in Northeast China among four wind erosion models and further evaluated model results using limited observations. The Method section gave a detailed description of the implementation of the four models, which can benefit the wind erosion modelers in terms of getting better understanding of the differences among the four models. The results of this study are supported by their analysis and very interesting. My main suggestion is performing a deeper analysis on the decreasing trend in the potential wind erosion as shown in Figure 3. Two things can be done regarding this suggestion: 1) using observations or satellite retrievals to cross-validate the simulated decreasing trend and 2) figuring out what factors drive the decreasing trend. This would be one of major findings of this study. One more suggestion is to compare the models’ sensitivities of the potential wind erosion to different factors, such as wind, land type, and vegetation coverage. Overall, I think this study could be an important contribution to the wind erosion community.

1) using observations or satellite retrievals to cross-validate the simulated decreasing trend.

Reply: Thanks for your insightful suggestion. Satellite retrievals is an excellent method to cross-validate the simulated decreasing trend, although it is not the real condition of potential wind erosion, but it still can be used for wind erosion trend (spatial or temporal) verification. However, the primary problem to validate wind erosion model is that there is no long-term standardized measured wind erosion data. Therefore, a long-term standard wind erosion monitoring network is urgently required in future research.

2) figuring out what factors drive the decreasing trend. This would be one of major findings of this study.

Reply: Thanks for your insightful suggestion. In fact, the authors have investigated that the relationship between annual wind erosion and the possible driving factors including wind speed, vegetation, soil moisture etc. Unfortunately, we did not find the obvious dominant factors in driving the decreasing trend of soil wind erosion for the studied region. However, some previous studies revealed that the driving force of soil wind erosion are the average vegetation coverage and average soil moisture (Du et al, 2017; Shen et al, 2016). As your suggestion, the authors intended to furthur examine the factors driving the decreasing trend in the future.

3) One more suggestion is to compare the models’ sensitivities of the potential wind erosion to different factors, such as wind, land type, and vegetation coverage.

Comments:

Reply: Thanks for your constructive and important suggestion. The authors have investigated the sensitivity of inputs of RWEQ and IWEMS (Zhang et al., 2022). Some previous studies also test the sensitivity of inputs of NWESMC and WEPS (Hagen et al,1999; Gao et al, 2012). These results concluded that wind speed was the most sensitive input parameter, and vegetation and soil moisture can control the wind erosion for different models. Thereby, here we investigated the spatial correlation analysis between potential wind erosion and wind speed, soil moisture, vegetation, precipitation and temperature (see the Table 4).

  1. In Abstract the four acronyms should be defined in their first appearance.

Reply: Thanks for your careful suggestion. The acronyms had been defined in Abstract.

  1. Line 35: should be dust emissions.

Reply: The “dust emission”. The authors have updated the word “dust emission” as “dust emissions” in the Revised Main Manuscript.

  1. Line 80–81: There’s no need to mention the names of these provinces since they are not marked in Figure 1.

Reply: Thanks for your kind suggestion. The names of those provinces had been deleted.

  1. Figure 1: Please use precipitation map to replace the topography map, because the APEC is defined closely based on precipitation rather than topography. Authors may add another panel in Figure 1 to include land use types.

Reply: Thanks for your careful suggestion. The authors have used the precipitation map to replace the topography map.

  1. Figure 3: Please use lines of different colors to represent different model results.

Reply: Thanks for your suggestion. Figure 3 has used use lines of different colors to represent different model results.

  1. Trend in Section 3.3: How the trends are estimated? Do they pass the statistically significant test?

Reply: This is an important question. The trend was estimated by the SPSS and the potential wind erosion presented a significant linear decline trend from 2000 to 2012. The interannual trend of four model passed the statistically significant test.

  1. The simulated decreasing trend in potential wind erosion should be cross-validated using observation and/or satellite data. For example, PM10 data and satellite retrieved AOD can be used. What factors drive the decreasing trend should be also investigated, such as winds, land use type, precipitation, vegetation, and so on?

Reply: Thanks for your insightful suggestion. Satellite retrievals or retrieved AOD is an excellent method to cross-validate the simulated decreasing trend, although it is not the real condition of potential wind erosion, but it still can be used for wind erosion trend (spatial or temporal) verification. As your suggestion, the PM10 data can be available for some sparse monitoring stations. It may be a powerful tool to verify the dust model (modeling the suspension). In summary, the primary problem to validate wind erosion models (modeling the saltation) is that there is no long-term standardized measured wind erosion data. Therefore, a long-term standard wind erosion monitoring network is urgently required in future research.

  1. Discuss the potential role of irrigation in the wind erosion in this region.

Reply: Thanks for your insightful suggestion. Soil moisture is an important factor governing wind erosion, and the soil moisture is determined by the natural precipitation and irrigation in the studied area. Here the data of irrigation is not available, thus we estimated potential wind erosion using soil moisture data.

  1. Figure 6: Most of the observational data points are located near zero, so I suggest use the log-scale for both x- and y-axis to zoom in the area where data density is high.

Reply: Thanks for your careful suggestion. The authors have used the log-scale for both x- and y-axis to present the data in the Figure 7.

Reviewer 2 Report

Dear Authors,

After a detailed review, it can be concluded that the manuscript is not well-written nor structured to international standard.

The manuscript aims to use four wind erosion calculation models to quantify the amount of wind erosion differences of spatial and temporal distribution in the study area. However, the length of the manuscript is too large and the cited reference are too much. It should be rewritten it concisely for the clarity of reader. In the whole manuscript structure, the advantages, disadvantages and characteristics of various wind erosion methods should be compared first to facilitate the reliability of the study according to one single and independent dataset. Then, that the schematic layout of the studied procedure should be illustrated to easily explain the significance of each step and the problems related to previous wind erosion studies. Finally, this conclusion must be rewritten to summarize the important conclusions of the studied results. For example, the results of four wind erosion models should be used to point out which wind erosion model is more suitable for the study area, and how to improve the estimated spatial and temporal accuracy in the future.

In my opinion, the manuscript should be rejected at this stage and the manuscript needs to be rewritten significantly to enhance its scientific value including the above the comparative results. The major comments are found in the attached PDF highlighted in yellow.

With my regards..

Author Response

After a detailed review, it can be concluded that the manuscript is not well-written nor structured to international standard.

The manuscript aims to use four wind erosion calculation models to quantify the amount of wind erosion differences of spatial and temporal distribution in the study area. However, the length of the manuscript is too large and the cited reference are too much. It should be rewritten it concisely for the clarity of reader. In the whole manuscript structure, the advantages, disadvantages and characteristics of various wind erosion methods should be compared first to facilitate the reliability of the study according to one single and independent dataset. Then, that the schematic layout of the studied procedure should be illustrated to easily explain the significance of each step and the problems related to previous wind erosion studies. Finally, this conclusion must be rewritten to summarize the important conclusions of the studied results. For example, the results of four wind erosion models should be used to point out which wind erosion model is more suitable for the study area, and how to improve the estimated spatial and temporal accuracy in the future.

In my opinion, the manuscript should be rejected at this stage and the manuscript needs to be rewritten significantly to enhance its scientific value including the above the comparative results. The major comments are found in the attached PDF highlighted in yellow.

Reply: The authors thank for your careful suggestions. This study provides the first insights into the difference of predicted regional wind erosion between these popular wind erosion models. Therefore, the authors used more detail materials to introduce the four modes and to describe the estimated results. Accordingly, we referred more cited references. We have updated and further polish the related sentences as your suggestions. (1) we rewrite the abstract; (2) the purposes of this study were updated; (3) we added the flow chart of the study; (4) we updated Table 1, Figure 4, Figure 7.

L21 showed

Reply: The “demonstrated”. The authors have updated the word “demonstrated” as “showed” in Line 21 in the Revised Main Manuscript.

L21 what is potential wind erosion? please define it!!

Reply: Thanks for your insightful question. Potential wind erosion means the maximum wind erosion modulus in the typical field of 100 m ´100 m in this paper.

L22 what is four models? not clear!!

Reply: Thanks for your question. The four models are NWESMC, RWEQ, WEPS and IWEMS, respectively. All of them had been showed in paper in red.

L23 published literature? not clear!!

Reply: Thanks for your kind question. The published literatures of observed wind erosion are all listed in Table 3.

L29 why? how does it cause any damages? please describe it!!

Reply: Thanks for your comment. Wind erosion model is generally developed based on specific target region, it may be not suitable to directly extend these models to other regions. In this case, the long-term standard measured wind erosion data can be used to calibrate and validate these models, and further greatly improve the accuracy of the model simulation. Therefore, it is urgent to establish a standardized long-term network for detecting soil wind erosion.

L64-71 this section should be rewritten. This study should strengthen the the purpose and focus on the comparative results from the adopted different wind erosion models and the measured values.

Reply: Thanks for your careful suggestion. Authors reinforce the purpose and highlight the results in different models in his section, and modify to the following: (1) to estimate the spatial-temporal trends of potential wind erosion using the regional versions of RWEQ and WEPS together with IWEMS and NWESMC in the APEC, (2) further validate the evaluated potential wind erosion using observed wind erosion data, and (3) to investigate the main factors affecting the regional potential wind erosion.

L73 Please draw the flow chart of this study, and then explain the significance of each step

Reply: Figure 2 is the flow chart of this study.

L75 and the problems related to previous wind erosion to be solved. what is APEC? Authors should explain it in detailed.

Reply: Thanks for your question, APEC is an abbreviation of the Agro-Pastoral Ecotone of northern China, and which had been modified in Line 17.

L88 map

Reply: The authors have updated the word “Schematic diagram” as “Schematic map” in the Revised Main Manuscript.

L103-105 These texts are too redundant. It is recommended to rearrange the relevant parameters into a list according the wind erosion formula

Reply: thanks for your suggestion. To unify the manuscript style, we have updated the content in this part.

L114-115 Not clear!! please rewrite all of them.

Reply: In NWESMC, potential wind erosion for each half a month was computed, and the annual potential wind erosion was calculated by the 24 half-monthly potential wind erosion.

L127 COG?

Reply: Sorry for misleading of parameter of COG, the incorrect section has been modified by the authors.

L254 these sections are messy. Please list each data source, calculation method, and purpose of application

Reply: Thanks for your careful suggestion. Each data of type, resolution, format and web sites have listed in table 1, and the calculation method and purpose of application were introduced in detailed in section 2.2.

L273 Please indicate the time of production of the data source

Reply: Thanks for your insightful question. In general, physical and chemical properties of soil and the undulating height of the earth’s surface are difficult to change greatly in a short period. The soil data and DEM we obtained can represent the condition of soil features and surface for a long period in study area. So these data have no temporal resolution.

L284-287 why? please explain all of the results

Reply: Thanks for your important question. According to the spatial distribution of soil wind erosion with different hazards erosion in Figure 3, the hazard of IWEMS is severely wind erosion, and the annual average number of soil wind erosion modes is higher than that of other model simulations.

L339-340 all the r2 squared value are too low. this means all of the wind erosion may be not used.

Reply: Thanks for your careful suggestion. The R2 values are low, and the pattern of the plot indicted that these models could not well predict the low potential wind erosion but only well estimate the big potential wind erosion. However, the R2 can still indicate the significant relationship between simulated and predicted wind erosion data.

L374-379 how to get R and P values?

Reply: Thanks for your careful question, the P values and R2 was obtained by SPSS.

L430 This conclusion must be rewritten to summarize the important conclusions of the studied results. For example, the results of four wind erosion models should be used to point out which wind erosion model is more suitable for the study area, and how to improve the estimated spatial and temporal accuracy in the future.

Reply: Thanks for your suggestion. The conclusions have rewritten by authors according to your recommendations.

Reviewer 3 Report

In this study, the National Wind Erosion Survey Model of China, the 18 Integrated Wind Erosion Modeling System, and the regional versions of the Revised Wind Erosion 19 Equation and Wind Erosion Prediction System were used to evaluate the regional potential wind 20 erosion of the agro-pastoral ecotone of northern China (APEC) in China during 2000 and 2012.  Results of the study may have important application in the field of soil conservation and management. Authors may wish to consider the following in revisions of their manuscript.

1.       Please discuss in more details of errors for the proposed models.

2.       Please comment on the limitations of using published data in comparison of model study, instead of using data actually collected by authors.

3.       What are the limitations of the proposed models.

4.       Please compare results obtained using proposed models with other similar models reported in the literature.

5.       Please have detailed discussions regarding use of the results of study to management of soil and soil conservation.

6.       Please include economic loss related to wind soil erosion in your paper.

Author Response

Reviewer 3

In this study, the National Wind Erosion Survey Model of China, the Integrated Wind Erosion Modeling System, and the regional versions of the Revised Wind Erosion Equation and Wind Erosion Prediction System were used to evaluate the regional potential wind erosion of the Agro-Pastoral Ecotone of northern China (APEC) during 2000 and 2012.  Results of the study may have important application in the field of soil conservation and management. Authors may wish to consider the following in revisions of their manuscript.

  1. Please discuss in more details of errors for the proposed models.

Reply: Thanks for your insightful question. The initial error for the proposed models is that we don’t have enough measured data to validate the models in our study area. And the wind erosion models including RWEQ, WEPS and IWEMS were developed based on different target regions.

  1. Please comment on the limitations of using published data in comparison of model study, instead of using data actually collected by authors.

Reply: Thanks for your suggestion. As you know, the territory of the APEC is vast, the authors do not have enough observed wind erosion data to calibrate and validate the four wind erosion models. Thus we collected to many observed wind erosion data from the published documents. Inevitably, there are several limitations for these wind erosion data. At first, the observed wind erosion data were obtained from different methods, such as radioisotope 137Cs method, sediment analysis and sand trap etc. Meanwhile, the sampling periods for different published data are also inconsistent. A long-term standard wind erosion monitoring network is urgently required.

  1. What are the limitations of the proposed models.

Reply: Thanks for your comments. Generally, the wind erosion models were developed based on different target regions. For example, RWEQ and WEPS were developed based on The Great Plains of the United States. IWEMS was developed based on Australia. Therefore, the parameters in these models are also determined based on the target regions. Accordingly, it is necessary to calibrate and validate the wind erosion models before extending them to other regions.

  1. Please compare results obtained using proposed models with other similar models reported in the literature.

Reply: Thanks for your insightful suggestion. The authors compared the results obtained using proposed models with other similar models reported in the literature. We also updated related section. The results of the four models are basically in the same order of magnitude as the relevant studies.

  1. Please have detailed discussions regarding use of the results of study to management of soil and soil conservation.

Reply: Thanks for your insightful and interesting question. Here the results show that the wind speed, soil humidity, vegetation cover could affect the wind erosion. It is necessary to curb the wind erosion by reducing wind speed (building wind barrier, improving vegetation cover, etc.) and soil conservation measures (e.g. retaining stubble.).

  1. Please include economic loss related to wind soil erosion in your paper.

Reply: Thanks for your constructive suggestions. Indeed, wind erosion could result in severe land degradation and reducing land productivity, which further influences region economic status. Your insightful suggestion provided us a valuable thread to study more interesting issue related to wind erosion. Thanks a lot!

Round 2

Reviewer 1 Report

The authors addressed some of my comments but with other comments not well-addressed.

1) MODIS satellite AOD data covers a period of 2000 to present and should be used to estimate the aerosol trend in this region. 

2) All trends and correlations in the manuscript should come with p-values or statistical significance level. SSPS is not a method but a software to estimate the trends. Authors should describe the method they used for trend estimation and significant test.

I hope the authors can carefully address the above comments before it can be considered for publication.

Author Response

The authors addressed some of my comments but with other comments not well-addressed.

Reply: Thanks for your suggestions, the authors further updated the related sentences.

1) MODIS satellite AOD data covers a period of 2000 to present and should be used to estimate the aerosol trend in this region. 

Reply: Thanks for your careful comments. The MODIS-AOD data set from 2003 to 2010 was provided by the TGP group, Institute of Remote Sensing and Digital Earth, Chinese Academy of Sciences (http://www.tgp.ac.cn/). As you pointed, fig.8 presented the spatial distribution and interannual variation the MODIS satellite AOD data from 2003 to 2010. High AOD generally scattered in the Mu Us Sands, Horqin Sands and the Hunshan Dake Sands, which were similar to the spatial distribution of severve wind erosion (Fig. 3). And the interannual variations of AOD is also consistent with the trends of annual potential wind erosion obtained from the four models (Fig. 4).

2) All trends and correlations in the manuscript should come with p-values or statistical significance level. SSPS is not a method but a software to estimate the trends. Authors should describe the method they used for trend estimation and significant test.

I hope the authors can carefully address the above comments before it can be considered for publication.

Reply: Thanks for your insightful suggestion. All trends and correlations were added with the p-values in the manuscript, and the correlations were significant at the 0.05 level (2-tailed). The authors have updated related sentences.

Reviewer 2 Report

Dear Authors,

This revision has been improved significantly according to the reviewer’s comments. In conclusion, this reviewer suggests that the manuscript is ready to be published in its present form.

With my regards.

Author Response

The authors thank you for your kindly and careful comments.